# Probiotic Properties of Lactic Acid Bacteria Isolated from the Spontaneously Fermented Soybean Foods of the Eastern Himalayas

Pynhunlang Kharnaior and Jyoti Prakash Tamang *

Department of Microbiology, School of Life Sciences, Sikkim University, Tadong, Gangtok 737102, Sikkim, India; kpynhun@gmail.com
* Correspondence: jptamang@cus.ac.in; Tel.: +91-9832-061-073

**Abstract:** Spontaneously fermented soybean foods with sticky-textured and umami-flavor are popular delicacies of multi-ethnic communities of the Eastern Himalayas. Even though species of *Bacillus* have been reported earlier as pre-dominant bacteria, we hypothesized that some lactic acid bacteria (LAB) present in these unique soy-based foods may exhibit probiotic properties. Hence, the present study is aimed to evaluate some probiotic attributes of LAB. A total of 352 bacterial isolates from spontaneously fermented soybean foods of the Eastern Himalayas viz., *kinema, grep-chhurpi, peha, peron namsing* and *peruñyaan* were preliminarily screened for survival in low pH, bile salt tolerance, and cell surface hydrophobicity. Finally, eight probiotic LAB were selected and identified, based on the 16S rRNA gene sequencing, as *Pediococcus acidilactici* Ki20 and *Enterococcus faecium* Kn19 (isolated from *kinema*), *E. faecalis* Gc21 (*grep-chhurpi*), *P. acidilactici* Ph32 (*peha*), *E. faecium* Pn11 and *E. faecalis* Pn37 (*peron namsing*), *E. lactis* Py03, and *E. lactis* Py08 (*peruñyaan*). In vitro probiotic attributes, *E. faecium* Kn19 (73.67 ± 1.05) and *P. acidilactici* Ph32 (79.71 ± 0.13%) recorded higher survival ability in acid and bile salt test, respectively. Furthermore, attachment ability of isolates to hydrocarbons showed ≥80% adhesion property with *E. faecalis* Gc21 (90.50 ± 10.14%) marked the highest degree of hydrophobicity, and *P. acidilactici* Ki20 showed the higher auto-aggregation and co-aggregation property. LAB strains were able to produce antibacterial activity against pathogenic bacteria. Genetic screening revealed the presence of genes responsible for acid tolerance (*groEl, clpL*), bile salt tolerance (*apf, bsh*), adhesion (*msa, mub1*), and bacteriocin producing genes for pediocin (*pedA, pedB*) and enterocin (*entA, entB*). The present study highlighted the probiotic potentials of LAB strains isolated from Himalayan naturally fermented soybean foods that may be developed as a starter or co-starter culture for controlled and optimized fermentation of soybeans.

**Keywords:** lactic acid bacteria; probiotic; *kinema*; spontaneously fermented soybeans; *Pediococcus*; Enterococcus





## 1. Introduction

Fermented foods have grown worldwide interest as functional foods, which enhanced and improved the availability of nutrients that gives additional value with health benefits to the consumers [1]. Several potential properties of fermented foods have increasing benefits due to the ability to enhance immunity, digestion, and lowering cholesterol levels [2]. In the Eastern Himalayas, numerous fermented products are available, and consumption of spontaneously fermented soybean foods is one of the robust dietary cultures of the Himalayan people living in Northeast India, Eastern Nepal, and Southern Bhutan [3]. Boiled soybean seeds are spontaneously fermented into sticky products with umami-flavor, which are known by different vernacular names in the Eastern Himalayas such as *kinema, tungrymbai, hawaijar, bekang, aakhonii/axone, bemerthu, grep-chhurpi, peha, peron namsing*, and *peruñyaan* [4–7]. Himalayan spontaneously fermented soybean foods harbour a diverse microbial community populated by bacteria with the dominance of *Bacillus subtilis* and another

species viz. *Bacillus velezensis, B. siamensis, B. tequilensis, B. safensis* subsp. *safensis, B. inaquosorum, B. halotolerans, B. glycinifermentans, B. cereus, B. licheniformis, B. thermoamylovorans, B. coagulans, B. circulans*, and *B. paralicheniformis* [8–13]. Species of *Bacillus* in the Himalayan fermented soybean foods exhibit several biological functions such as the production of poly-glutamic acid [12,14], the production of untargeted metabolites [11], anti-thrombolytic property [4,7], the production of bio-peptides [15], and the enhancement of antioxidant properties [16,17].

Since soybeans are spontaneously fermented, the occurrence of a few species of lactic acid bacteria (LAB) in lower abundances during fermentation has also been reported, which included *Enterococcus faecium, Lactiplantibacillus plantarum*, and *Limosilactobacillus fermentum* in *kinema* [8,10,18–20]. LAB in fermented soybean products have been reported to play a crucial role during fermentation due to their antibacterial properties [21], production of protease [22], an increase in isoflavone content, antioxidant capacity [23], and their probiotic properties [24]. Several LAB species were reported with probiotic properties such as *Lactococcus lactis* from *doenjang*, Korean fermented soybean paste [25]; *Lactobacillus rhamnosus* and *Lactobacillus casei* from *douchi,* Chinese fermented soybean paste [26]; *Leuconostoc mesenteroides, Lactiplantibacillus plantarum, Levilactobacillus brevis* and *Lactobacillus perolens* from *cheonggukjang,* Korean fermented soybean food [27]; and *Lactiplantibacillus plantarum* from *kinema,* Himalayan fermented soybean food [20].

However, the probiotic property of LAB species in the Himalayan fermented soybean has not been assessed yet, except *Lactiplantibacillus plantarum* from *kinema* [20]. Hence, the present study is aimed to screen and identify the lactic acid bacteria in some spontaneously fermented soybean foods of the Eastern Himalayas viz. *kinema, grep-chhurpi, peha, peron namsing,* and *peruñyaan*, using 16S rRNA gene sequencing and evaluation of some probiotic attributes of isolates by in-vitro and genetic screening based on marker gene of certain characteristics.

## 2. Materials and Methods

### 2.1. Collection of Sample

A total of 52 samples of naturally fermented soybean foods were aseptically collected from different regions of the Eastern Himalayas. Six sample of *kinema* from Dharan, Nepal (Altitude = 371 m; Latitude = 26.8065° N; Longitude = 87.2846° E) and Ilam, Nepal (Altitude = 1205 m; Latitude = 26.9112° N; Longitude = 87.9237° E), six samples of *kinema* from Darjeeling hills, India (Altitude = 2042 m; Latitude = 27.0410° N; Longitude = 88.2663° E) and Sikkim, India (Altitude = 1650 m; Latitude = 27.3314° N; Longitude = 88.6138° E), six samples of *kinema* from Bhutan (Altitude = 417 m; Latitude = 26.9131° N; Longitude = 89.0836° E), four samples of *grep-chhurpi* from Tawang, Arunachal Pradesh in India (Altitude = 3048 m; Latitude = 27.5861° N; Longitude = 91.8594° E), ten samples of *peha* from Itanagar, Arunachal Pradesh (Altitude = 320 m; Latitude = 27.1719° N; Longitude = 93.7029° E), fourteen samples of *peron namsing* from Pasighat, Arunachal Pradesh (Altitude = 152 m; Latitude = 28.0632° N; Longitude = 95.3239° E), and six samples of *peruñyaan* from Ziro, Arunachal Pradesh (Altitude = 5538 m; Latitude = 27.6169° N; Longitude = 93.8392° E). The samples were collected in pre-sterile containers, sealed, labelled, and transported to the laboratory in an ice-box cooler, and stored in a deep freezer for subsequent analysis.

### 2.2. Microbiological Analysis

Ten grams of samples mixed with 90 mL of pre-sterilized physiological saline (0.85% NaCl) were homogenized using a stomacher blender (400, Seward, London, UK) and performed for a serial dilution ($10^{-1}$ to $10^{-8}$). The sample mixture (1 mL) of each dilution was transferred to MRS (de Man-Rogosa-Sharpe) agar plates (GM641, HiMedia, Mumbai, India) containing 1% $CaCO_3$ (GRM1044, HiMedia, Mumbai, India) and incubated under anaerobic condition for 24–48 h at 30 °C. After incubation, colonies were purified and proceeded for further analysis [28].

*2.3. Preliminary Screening of Probiotic Isolates*

2.3.1. Acid Tolerance Test

The overnight-grown isolates in MRS broth (GM369, HiMedia, Mumbai, India) were centrifuged at 8000× *g* for 5 min at 4 °C. The MRS broth was adjusted at pH 3.0 with 1N HCl [29] and the harvested cell pellets of isolates were resuspended in MRS broth with acidic pH and incubated at 30 °C for 24 h. The un-inoculated MRS broth served as control. The cell density of the isolates in MRS broth with acidic pH was observed at 600 nm using an Eppendorf BioSpectrometer (Model No. 6135 000 009, Hamburg, Germany). The minimum thresholds of ≥0.5 optical density for the survival of isolates in low pH were selected for further probiotic assessment [30].

2.3.2. Bile Salt Tolerance Test

Bile salt tolerance of isolates was carried out following the method described by Bao et al. [30]. Isolates inoculated in pre-sterilized MRS broth were incubated at 30 °C for 24 h. A cell density of 2% (±0.1) was transferred into MRS broth containing 0.3% ox-bile (CR010, HiMedia, Mumbai, India) with un-inoculated broth served as control, followed by incubation for 24 h at 30 °C. After incubation, a cell density was observed at 600 nm using an Eppendorf BioSpectrometer (Model No. 6135 000 009, Hamburg, Germany). The minimum thresholds of ≥0.5 optical density for the survival of isolates in bile salts were selected for further probiotic assessment [30].

2.3.3. Assessment of Cell Surface Hydrophobicity

The adhesive ability of isolates to the epithelial cells was performed using the method of Nath et al. [31]. A 24 h fresh culture broth was centrifuged at 8000× *g* for 5 min at 4 °C, and cell pellets were collected and washed twice using Phosphate Buffered Saline (PBS) buffer (M1452, HiMedia, Mumbai, India). Cell pellets were re-suspended in PBS buffer and measured the optical density at 600 nm using an Eppendorf BioSpectrometer (Model No. 6135 000 009, Hamburg, Germany) maintaining 1.0 as initial absorbance ($A_{Initial}$). Thereafter, 3 mL of cell suspensions were mixed with 1 mL of hydrocarbons [n-hexadecane (RM2238, HiMedia, Mumbai, India) and toluene (AS072, HiMedia, Mumbai, India)] and vortexed for 2 min, followed by incubation at room temperature and kept undisturbed for 1 h. After incubation, the aqueous phase was removed to measure the final absorbance ($A_{Final}$) at 600 nm. The cell surface hydrophobicity rate was calculated using the formula:

$$\text{Cell Surface Hydrophobicity (\%)} = [(A_{Initial} - A_{Final})/A_{Final}] \times 100$$

*2.4. Genotypic Identification*

2.4.1. Genomic DNA Extraction

The genomic DNA of isolates was extracted following the method described by Shangpliang and Tamang [28]. Briefly, an 18–24 h fresh culture was transferred into a 2 mL microcentrifuge tube and centrifuged at 8000× *g* for 5 min. The cell pellet was then collected and washed twice with sterile 0.5 M sodium chloride (GRM031, HiMedia, Mumbai, India), followed by immediate rinsing with sterile deionized water. Further, the cell pellet was resuspended with 500 µL of 1X TE buffer (pH 8), followed by the addition of 10-µL lysozyme (20 mg/mL) (MB098, HiMedia, Mumbai, India) to the solution. Additionally, the cell suspension was incubated for 30 min at 37 °C for enzyme activation followed by heating at 98 °C for 15 min. The supernatant was collected in a sterile microcentrifuge tube after centrifugation at 10,000× *g* for 10 min at 4 °C. Lastly, DNA was quantified using Eppendorf BioSpectrometer (Model No. 6135 000 009, Hamburg, Germany). DNA purity with an absorbance ($A_{260}/A_{280}$) of 1.8 to 2.2 was carried out for polymerase chain reaction (PCR).

### 2.4.2. PCR Amplification

A 50-µL volume of PCR amplification was performed using a reaction containing GoTaq® Green Master Mix (M7122, Promega, WI, USA), 30–50 ng of DNA template and 27F 5′-AGAGTT TGATCATGGCTCAG-3′; 1492R 5′-GTTACCTTGTTA CGACTT-3′ [32]. Veriti^TM Thermal cycler (4375305, Applied Biosystems, Thermo Fisher Scientific, Carlsbad, CA, USA) was used for PCR amplification using the following conditions: initial denaturation of PCR amplification (94 °C for 5 min), followed by 30 cycles denaturation (94 °C for 1 min), annealing process (55 °C for 1 min) and elongation (72 °C for 1.5 min), respectively. Lastly, PCR amplification was set for continuation of elongation at 72 °C for up to 10 min followed by a stoppage at 4 °C.

### 2.4.3. Purification of PCR

Amplified PCR products were purified following the method described by Shangpliang and Tamang [28]. PCR amplicons were mixed with 0.6 volumes of polyethylene glycol-sodium chloride (PEG-NaCl) and incubated at 37 °C for 30 min, followed by centrifugation at 10,000× *g* for 30 min at 4 °C. The DNA (cell pellet) was collected, washed twice with 70% freshly prepared ethanol, and air dried. Finally, the purified DNA pellet was resuspended with 30 µL nuclease-free water and checked the quality by agarose gel electrophoresis (0.8%) and visualized using Gel Doc™ EZ Imager (Model No. EZ735BR06435, BioRad, Hercules, CA, USA).

### 2.4.4. 16S rRNA Gene Sequencing

The purified PCR amplicon was prepared for sequencing library preparation. A set of primers including both 27F 5′-AGAGTTTGATCATGGCTCAG-3′ forward and 1492R 5′-GTTACCTTGTTACGACTT-3′ reverse primers [32] were used and prepared with two sets of separate sequencing reaction for forward and reverse primers, respectively. A final reaction volume of 50 µL containing 0.2 µM primer, 0.2 mM dNTPs (dATPs, dTTPs, dGTPs, dCTPs), 2.0 mM MgCl$_2$, 0.5 mg/mL, and 0.04 U/µL Taq DNA polymerase. Further, the PCR reaction was prepared under the following conditions initiated by initial denaturation (95 °C for 10 min), denaturation process for 35 cycles (95 °C for 1 min), annealing process (40 °C for 2 min), elongation (72 °C for 1 min) and further elongation process at 72 °C for 10 min. Lastly, an automated DNA analyzer (ABI 3730XL Capillary Sequencers, Applied Biosystems, Foster City, CA, USA) was used for sequencing the library-prepared DNA.

### *2.5. In Vitro Screening of Probiotic Properties*

### 2.5.1. Survival to Acid and Bile Salt

The acid and bile survival properties of isolates were determined by following the method described by Mallappa et al. [33]. The culture was harvested by centrifugation at 8000× *g* for 5 min at 4 °C. The cell pellets were resuspended in pre-adjusted MRS broth (pH 3) and 0.3% bile salt, respectively, followed by incubation for 3 h at 37 °C. The samples before and after incubation were plated on MRS agar and incubated at 37 °C for 48 h to determine the survival rate after exposure.

### 2.5.2. Auto-Aggregation and Co-Aggregation Assays

Auto-aggregation and co-aggregation of isolates were determined following the method described by Li et al. [34]. In auto-aggregation, overnight culture incubated at 37 °C using MRS broth was harvested by centrifugation at 5000× *g* for 15 min. The cell pellets were washed three times using PBS buffer (pH 7.2) and resuspended in 2 mL PBS solution adjusting the cell density to 0.1 ± 0.05 at 600 nm using an Eppendorf BioSpectrometer (Model No. 6135 000 009, Hamburg, Germany) and further incubated at 37 °C for 3 h. After incubation, 100 µL of the upper part of the bacterial cell suspension was collected and

observed the final absorbance ($OD_{A600}$). The auto-aggregation percentage was determined using the following equation:

$$\text{Auto-aggregation (\%)} = [(A_{Initial} - A_{Final})/A_{Initial}] \times 100$$

where, $A_{Initial}$ denotes the absorbance at time = 0, and $A_{Final}$ denotes the absorbance at time = 3 h.

For co-aggregation, isolates were tested for their ability to adhere to pathogenic strains such as *Escherichia coli* KL96 MTCC (Microbial Type Culture Collection, Chandigarh, India) 1583, *Salmonella enterica* subsp. *enterica* ser. *typhimurium* MTCC 3223, *Staphylococcus aureus* subsp. *aureus* MTCC 740, and *Bacillus cereus* MTCC 1272. Briefly, an overnight freshly prepared bacterial cell suspension was centrifuged at $5000 \times g$ for 15 min for both LAB and pathogens. The cell pellets were collected, washed three times with PBS buffer, resuspended in 2 mL (PBS), and adjusted to an optical density of $0.1 \pm 0.05$ at 600 nm using an Eppendorf BioSpectrometer (Model No. 6135 000 009, Hamburg, Germany). The cell suspensions of an equal volume of isolates and pathogenic strains were mixed and incubated at 37 °C for 3 h. After incubation, the absorbance of the bacterial cell suspension mixture was determined at 600 nm and the co-aggregation rate (%) was calculated as follows:

$$\text{Co-aggregation percentage (\%)} = [(A_{LAB} + A_{Pathogen}) - A_{Mix}]/(A_{LAB} + A_{Pathogen}) \times 100$$

### 2.5.3. Resistance to Lysozyme

The resistivity of isolates against lysozyme was performed following the method of Vera-Pingitore et al. [35]. An overnight fresh culture was harvested by centrifugation at $8000 \times g$ for 10 min to obtain the cell pellets and washed by PBS buffer. Harvested cell pellets were resuspended with 5 mL (PBS) and measured the cell density for initial absorbance ($A_{Initial}$) at 600 nm using Eppendorf BioSpectrometer (Model No. 6135 000 009, Hamburg, Germany). One mL of bacterial suspension was inoculated into the solution containing 100 µg/mL of lysozyme (MB098, HiMedia, Mumbai, India) and incubated at 37 °C for 1 h. After the incubation period, cell density was measured for final absorbance ($A_{Final}$) at 600 nm and calculated the resistance percentage (%) as follows:

$$\text{Resistance to lysozyme (\%)} = (A_{Final}/A_{Initial}) \times 100$$

### 2.5.4. Bile Salt Hydrolase (BSH) Activity

BSH activity of the isolate was performed following the method described by Pradhan and Tamang [36]. A minor modification of MRS agar supplemented with 0.5% (*w/v*) sodium taurocholate (RM011, HiMedia, Mumbai, India), sodium taurodeoxycholate (TC347, HiMedia, Mumbai, India), and 0.37 g/L of $CaCl_2$ (GRM399, HiMedia, Mumbai, India) was prepared. Isolates were streaked on the modified MRS agar and incubated anaerobically at 37 °C for 72 h. After incubation, BSH activity was demonstrated by the presence of precipitated bile acid around the colonies showing clear zones.

### 2.5.5. Antagonistic Activity

Antimicrobial activity of isolates against pathogenic bacteria such as *Escherichia coli* MTCC 1583, *Salmonella enterica* subsp. *enterica* ser. *typhimurium* MTCC 3223, *Staphylococcus aureus* subsp. *aureus* MTCC 740, and *Bacillus cereus* MTCC 1272 was conducted following the method described by Rai and Tamang [37]. Briefly, an overnight freshly prepared tested pathogen was adjusted to a density of 0.08 to 0.1 ($\pm 0.05$) and spread on the surface of Mueller Hinton agar (M173, HiMedia, Mumbai, India) plates by sterile cotton swab and wells punching, made by a sterile borer. Overnight LAB cultures were inoculated into the wells and incubated at 37 °C for 24–48 h. After incubation, the zone of inhibition was observed.

### 2.5.6. Genetic Screening for Probiotic Functions

Gene detection of various probiotic functions was screened for isolates using a PCR-based method [38–45]. Details of genes and primers with probiotic function including bile salt tolerance, low pH tolerance, adherence/attachment, and bacteriocin production are listed in Table 1. For gene detection, a PCR reaction was prepared with a total volume of 10-μL containing GoTaq® Green Master Mix (M7122, Promega, WI, USA), DNA template, and primers (forward and reverse). Targeted gene amplification was performed using Veriti™ Thermal cycler (4375305, Applied Biosystems, Thermo Fisher Scientific, Carlsbad, CA, USA) under the following conditions: denaturation of PCR amplification (94 °C for 5 min), followed by 30 cycles denaturation (94 °C for 1 min), annealing process (55 °C for 1 min) and elongation process (72 °C for 1.5 min), respectively. Lastly, PCR amplification was set up for the continuation of the elongation process (72 °C for 10 min) and a stoppage process at 4 °C. The amplified PCR products were checked by agarose gel electrophoresis (0.8%) and visualized using Gel Doc™ EZ Imager (EZ735BR06435, BioRad, Hercules, CA, USA).

**Table 1.** Details of marker genes studied, and their respective primers used for detection of probiotic functions.

| Genes | Functions | Primer Sequence (5′ → 3′) (F = Forward; R = Reverse) | Annealing Temperature (°C) | Size of Amplicon (bp) | References |
|---|---|---|---|---|---|
| *groEl* | Survival at low pH | F-TTCCATGGCKTCAGCRATCA R-GCTAAYCCWGTTGGCATTCG | 58 | 168 | [38] |
| *clpL* | Survival at low pH | F-GCTGCCTTYAAAACATCATCTGG R-AATACAATTTTGAARAACGCAGCTT | 50 | 158 | [38] |
| *odc* | Survival at low pH | F-TMTWCCAACHGATCGWAATGC R-CRCCCCAWGCACARTCRAA | 52 | 245 | [38] |
| *tdc* | Survival at low pH | F-CCACTGCTGCATCTGTTTG R-CCRTARTCNGGNATAGCRAARTCNGTRTG | 50 | 370 | [38] |
| *Ir0085* | Bile salt | F-RCTTTGACCGRTGGGGCTRT R-NNNATGGCCGCATGGAAA | 57.5 | 150 | [38] |
| *Ir1516* | Bile salt | F-TRACCACTYTCWCCATTCAACAA R-CCACTAGCRATGACYAATACKGGT | 56.5 | 143 | [38] |
| *apf* | Bile salt | F-YAGCAACACGTTCTTGGTTAGCA R-GAATCTGGTGGTTCATAYWCAGC | 53 | 112 | [38] |
| *bsh* | Bile salt | F-ATTGAAGGCGGAACSGGMTA R-ATWACCGGWCGGAAAGCTG | 58 | 155 | [38] |
| *mub1* | Adhesion | F-GTAGTTACTCAGTGACGATCAATG R-TAATTGTAAAGGTATAATCGGAGG | 50 | 150 | [39] |
| *msa* | Adhesion | F-GCGATTAGGGGTGTGCAAG R-GCAGTTGGTGACGTAGGCA | 55 | 319 | [40] |
| *fbp* | Adhesion | F-AGTGCTGAAATYATGGGAAGA R-AATTGTCCACCTTGTTGCTG | 60 | 835 | [40] |
| *entA* | Bacteriocin | F-GGT ACC ACT CAT AGT GGA AA R-CCC TGG AAT TGC TCC ACC TAA | 55 | 138 | [41] |
| *entB* | Bacteriocin | F-CAA AAT GTA AAA GAA TTA AGT ACG R-AGA GTA TAC ATT TGC TAA CCC | 56 | 201 | [42] |
| *pedA* | Bacteriocin | F-AAAATATCTAACTAATACTTG R-TAAAAAGATATTTGACCAAAA | 44 | 600 | [43] |

| Genes | Functions | Primer Sequence (5′ → 3′) (F = Forward; R = Reverse) | Annealing Temperature (°C) | Size of Amplicon (bp) | References |
|---|---|---|---|---|---|
| *pedB* | Bacteriocin | F-ATGAATAAGACTAAGTCGGAACATATT R-CTATTGGCTAGGCCACGTATTG | 57 | 339 | [44] |
| *cylA* | Bacteriocin | F-ACTCGGGGATTGATAGGC R-GCTGCTAAAGCTGCGCTT | 54 | 688 | [45] |

### 2.6. Bioinformatics Analysis

The raw sequence generated by Sanger sequencing was evaluated for the quality assessment using Sequence Scanner (v2.0) software of Applied Biosystems (https://www.thermofisher.com/in/en/home/life-science/sequencing/sanger-sequencing/sanger-dna-sequencing/sanger-sequencing-data-analysis.html) accessed on 2 December 2023. Based on the quality trace score (Q > 20) and length (>600 bp), good-quality sequences were assembled using ChromasPro v2.1.10 (http://technelysium.com.au/wp/chromas/) accessed on 2 December 2022 and chimera sequence filtration checked by Mallard programme [46]. Further, the good-quality assembled sequences were identified by mapping against the NCBI database using BLAST (Basic Local Alignment Search Tool) [47]. Lastly, the identified lactic acid bacterial strains were aligned using ClustalW [48] and constructed the molecular evolutionary analysis by the Neighbor-joining method [49] using MEGA11.0.13 (Molecular Evolutionary Genetics Analysis) [50].

All the experiments on in-vitro probiotic attributes were conducted in triplicates and represented in Mean ± SD (standard deviation).

## 3. Results

### 3.1. Preliminary Screening

A total of 352 bacterial isolates from *kinema* (114 isolates), *grep-chhurpi* (38 isolates), *peha* (71 isolates), *peron namsing* (87 isolates), and *peruñyaan* (42 isolates) were preliminarily screened for probiotic properties on the basis of low pH and bile salt (0.3%) tolerances. Out of which, only 54 isolates showed tolerances to low acid and bile salt. Further, 54 isolates were again screened for cell surface hydrophobicity. Finally, eight isolates showing >80% hydrophobicity were tentatively selected as probiotic bacteria for in vitro screening of probiotic attributes. Based on 16S rRNA gene sequencing (Figure 1), eight isolates were identified as *Pediococcus acidilactici* Ki20 and *Enterococcus faecium* Kn19 (isolated from *kinema*), *Enterococcus faecalis* Gc21 (*grep-chhurpi*), *Pediococcus acidilactici* Ph32 (peha), *Enterococcus faecium* Pn11 and *Enterococcus faecalis* Pn37 (*peron namsing*), *Enterococcus lactis* Py03, and *Enterococcus lactis* Py08 (*peruñyaan*) (Table 2).

**Table 2.** Lactic acid bacteria isolated from naturally fermented soybean food of the Eastern Himalayas with NCBI accession numbers and reference type strains.

| Products | Identity with Sample Code | Type Species (% Similarity) | GenBank Accession Number |
|---|---|---|---|
| *Kinema* | *Pediococcus acidilactici* Ki20 | *Pediococcus acidilactici* DSM 20284 (99.72%) | OP941712 |
| | *Enterococcus faecium* Kn19 | *Enterococcus faecium* LMG 11423 (99.65%) | OP941713 |
| *Grep chhurpi* | *Enterococcus faecalis* Gc21 | *Enterococcus faecalis* ATCC 19433 (99.85%) | OP941714 |
| *Peha* | *Pediococcus acidilactici* Ph32 | *Pediococcus acidilactici* DSM 20284 (99.86%) | OP941715 |
| *Peron namsing* | *Enterococcus faecium* Pn11 | *Enterococcus faecium* LMG 11423 (99.52%) | OP941716 |
| | *Enterococcus faecalis* Pn37 | *Enterococcus faecalis* ATCC 19,433 (99.93%) | OP941717 |
| *Peruñyaan* | *Enterococcus lactis* Py03 | *Enterococcus lactis* BT159 (99.64%) | OP941718 |
| | *Enterococcus lactis* Py08 | *Enterococcus lactis* BT159 (99.71%) | OP941719 |

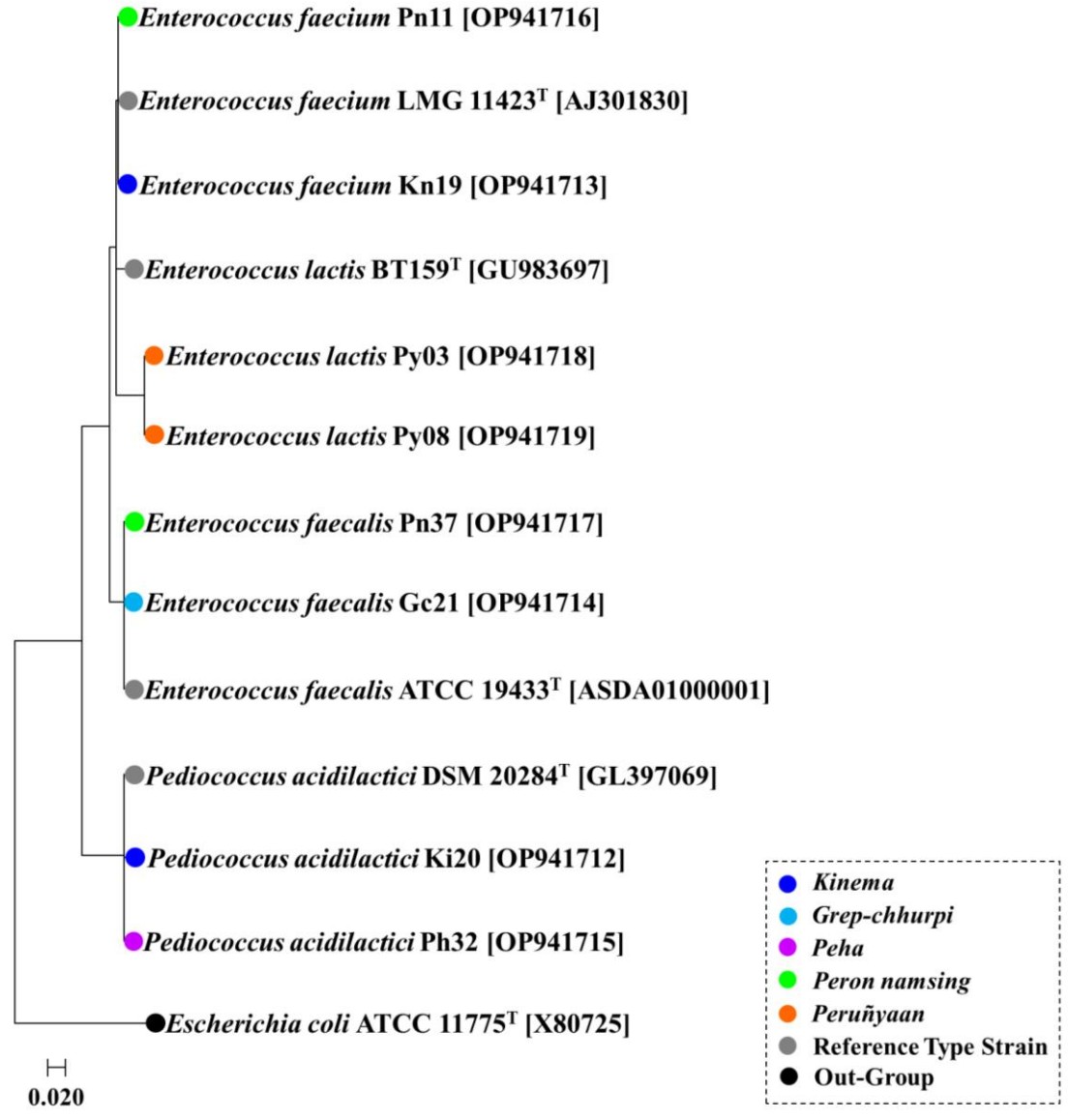

**Figure 1.** Phylogenetic analysis of lactic acid bacteria isolated from spontaneously fermented soybean foods of the Eastern Himalayas by the neighbor-joining method using MEGA11, based on 16S rRNA gene sequences. The bootstrap consensus of 1000 replicates was taken to represent the evolutionary history and Kimura-2 parameter method for evolutionary distances. *Pediococcus acidilactici* DSM 20284, *Enterococcus faecalis* ATCC 19433, *Enterococcus lactis* BT159, and *Enterococcus faecium* LMG 11423 were used as reference type strains, and *Escherichia coli* ATCC 11775 as an outgroup.

### 3.2. In-Vitro Probiotic Properties

In vitro probiotic properties of eight LAB, isolated from the Himalayan fermented soybean foods were assessed. The survival rate of tested LAB was found >52% in pH 3.0 and >66% in bile salt, among which *Enterococcus faecium* Kn19 (*kinema*) and *Pediococcus acidilactici* Ph32 (*peha*) showed the highest survival rate of 73.67 ± 1.05% in low pH and 79.71 ± 0.13% in 3% bile salt, respectively (Table 3). More than 85% of hydrophobicity percentage was observed in all eight LAB, among which *Enterococcus faecalis* Gc21 (*grep-chhurpi*) showed the highest hydrophobicity percentage of 90.50 ± 10.14% (Table 3). Auto-aggregation percentage of all tested LAB ranged from 43.8 ± 1.05 to 55.73 ± 0.96 (Table 3). Moreover, all LAB strains showed co-aggregation property against *Bacillus cereus* MTCC 1272, *Escherichia coli* MTCC 1583, *Salmonella enterica* subsp. *enterica* ser. *typhimurium* MTCC 3223, and *Staphylococcus aureus* subsp. *aureus* MTCC 740 (Table 3). *Pediococcus acidilactici*

Ki20 isolated from *kinema* showed the highest co-aggregation property against the tested pathogens (Table 3). Similarly, resistance to lysozyme was exhibited by all LAB, among which *Pediococcus acidilactici* Ph32 exhibited the highest resistance to lysozyme (Table 3). Further, in vitro probiotic attributes were tested for bile salt hydrolase activity (Table 3), in which five isolates were positive to sodium taurodeoxycholate and four isolates to Sodium taurocholate (Table 3). Additionally, the antagonistic activity of LAB, against four pathogens including *Bacillus cereus* MTCC 1272, *Escherichia coli* MTCC 1583, *Salmonella enterica* subsp. *enterica* ser. *typhimurium* MTCC 3223, and *Staphylococcus aureus* subsp. *aureus* MTCC 740, was performed and found. Only *Enterococcus faecium* Kn19 (*kinema*), *E. faecalis* Gc21 (*grep-chhurpi*), *Pediococcus acidilactici* Ph32 (*peha*), and *E. faecalis* Pn37 (*peron namsing*) exhibited antibacterial activity against all the tested pathogens, whereas other LAB strains showed the variable results (Figure 2).

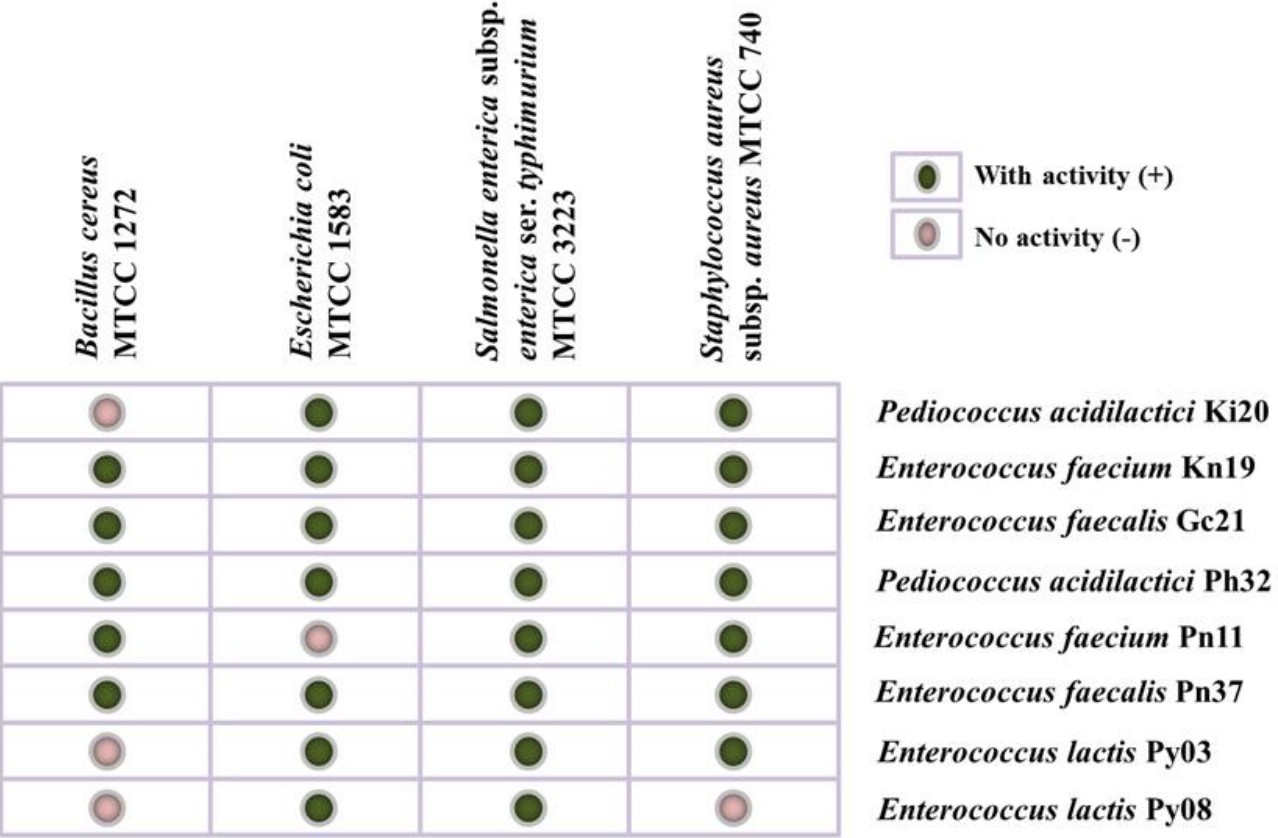

**Figure 2.** Antibacterial activity of identified lactic acid bacterial strains against some pathogenic bacteria.

**Table 3.** Probiotic properties of lactic acid bacterial strains isolated from spontaneously fermented soybean foods of the Eastern Himalayas.

| Samples | Lactic Acid Bacteria | Survival Rate (%) | | Cell Surface Hydrophobic-ity (%) | Auto-Aggregation (%) | Co-Aggregation (%) | | | | Resistance to Lysozyme (%) | BSH Activity | |
|---|---|---|---|---|---|---|---|---|---|---|---|---|
| | | Acid (pH 3) | Bile (0.3% Oxgall) | | | *Bacillus cereus* MTCC 1272 | *Escherichia coli* MTCC 1583 | *Salmonella enterica* subsp. *enterica* ser. *typhimurium* MTCC 3223 | *Staphylococcus aureus* subsp. *aureus* MTCC 740 | | Sodium Taurodeoxy-cholate | Sodium Tauro-cholate |
| Kinema | *Pediococcus acidilactici* Ki20 | 67.85 ± 0.25 | 78.03 ± 0.21 | 86.67 ± 3.30 | 55.73 ± 0.96 | 79.51 ± 0.35 | 72.20 ± 0.72 | 75.52 ± 0.34 | 75.14 ± 0.52 | 62.38 ± 0.59 | + | + |
| | *Enterococcus faecium* Kn19 | 73.67 ± 1.05 | 74.08 ± 0.38 | 87.84 ± 3.06 | 50.03 ± 1.14 | 65.02 ± 1.04 | 58.48 ± 1.77 | 58.52 ± 0.87 | 58.97 ± 1.13 | 70.80 ± 0.46 | + | + |
| Grep-chhurpi | *Enterococcus faecalis* Gc21 | 52.02 ± 1.05 | 66.46 ± 0.88 | 90.50 ± 10.14 | 54.15 ± 0.29 | 59.72 ± 0.93 | 56.31 ± 0.45 | 60.99 ± 0.78 | 59.06 ± 0.25 | 75.09 ± 0.77 | − | − |
| Peha | *Pediococcus acidilactici* Ph32 | 66.11 ± 2.32 | 79.71 ± 0.13 | 88.97 ± 6.55 | 43.80 ± 0.40 | 64.52 ± 0.25 | 58.58 ± 1.11 | 60.97 ± 0.64 | 60.82 ± 0.41 | 77.76 ± 0.25 | + | + |
| Peron namsing | *Enterococcus faecium* Pn11 | 55.92 ± 1.85 | 75.32 ± 0.95 | 89.34 ± 6.60 | 48.13 ± 0.33 | 59.13 ± 0.40 | 59.69 ± 0.39 | 66.41 ± 0.77 | 55.93 ± 0.79 | 66.08 ± 0.30 | − | + |
| | *Enterococcus faecalis* Pn37 | 61.43 ± 0.65 | 75.54 ± 0.55 | 85.67 ± 1.89 | 55.05 ± 0.36 | 63.66 ± 0.25 | 56.90 ± 0.42 | 59.58 ± 0.42 | 58.10 ± 1.03 | 67.78 ± 1.05 | − | − |
| Peruñyaan | *Enterococcus lactis* Py03 | 53.24 ± 1.17 | 66.41 ± 0.85 | 89.00 ± 11.31 | 55.46 ± 0.48 | 64.01 ± 0.92 | 51.29 ± 0.75 | 60.23 ± 0.50 | 58.54 ± 0.24 | 64.48 ± 1.75 | + | − |
| | *Enterococcus lactis* Py08 | 64.92 ± 1.60 | 71.88 ± 0.14 | 90.34 ± 8.96 | 54.10 ± 0.38 | 58.14 ± 0.27 | 52.69 ± 0.70 | 53.98 ± 0.26 | 54.41 ± 0.52 | 72.53 ± 1.39 | + | − |

Note: All experiments were conducted in triplicates represented in Mean ± SD. '+' = Activity; '−' = No activity; BSH = Bile Salt Hydrolase; MTCC = Microbial Type Culture Collection.

### 3.3. Gene Detection of Probiotic Functions

Genes responsible for various probiotic traits were screened for all eight LAB strains using a PCR-based method. Genetic screening of probiotic function for survival at low pH showed the presence of *groEl* in *Pediococcus acidilactici* Ki20 and *Enterococcus faecium* Kn19 (*kinema*), *E. faecalis* (*grep-chhurpi*), *P. acidilactici* Ph32 (*peha*), *E. lactis* Py03 and *E. lactis* Py08 (*peruñyaan*) (Figure 3a) and *clpL* genes in *P. acidilactici* Ki20 and *E. faecium* Kn19 (*kinema*), *P. acidilactici* Ph32 (*peha*), *E. faecium* Pn11 and *E. faecalis* Pn37 (*peron namsing*) and *E. lactis* Py08 (*peruñyaan*) (Figure 3b); however, none of the strains found to harbour *odc* and *tdc* genes. Further, genes responsible for bile salt tolerance were screened, and all strains showed the presence of *apf* (aggregation-promoting factor) gene (Figure 3c). Similarly, *bsh* gene responsible for bile salt hydrolase (BSH) was also detected in all strains (Figure 3d). All strains harboured the *msa* gene for cell surface adherence/attachment (Figure 3e), and only four strains viz. *E. faecalis* Gc21 (*grep-chhurpi*), *E. faecium* Pn11 and *E. faecalis* Pn37 (*peron namsing*), and *E. lactis* Py03 (*peruñyaan*) showed the presence of *mub1* gene (Figure 3f). *P. acidilactici* Ki20 (*kinema*) and *P. acidilactici* Ki20 (*peha*) were detected with genes *ped A* (Figure 4a) and *ped B* (Figure 4b), respectively, for pediocin. Gene *ent A* was detected in *E. faecium* Kn19 and Gc21 (*grep-chhurpi*), *E. faecium* Pn11 and *E. faecalis* Pn37 (*peron namsing*), and *E. lactis* Py03 (*peruñyaan*) (Figure 4c). Gene *ent B* was detected in *E. faecium* Kn19 (*kinema*), *E. faecalis* Gc21 (*grep-chhurpi*), *E. faecalis* Pn37 (*peron namsing*), and *E. lactis* Py08 (*peruñyaan*) (Figure 4d). None of these genes, *odc*, *tdc*, *Ir0085*, *Ir1516*, *fbp*, and *cylA*, were detected in LAB strains isolated from the Himalayan fermented soybean foods.

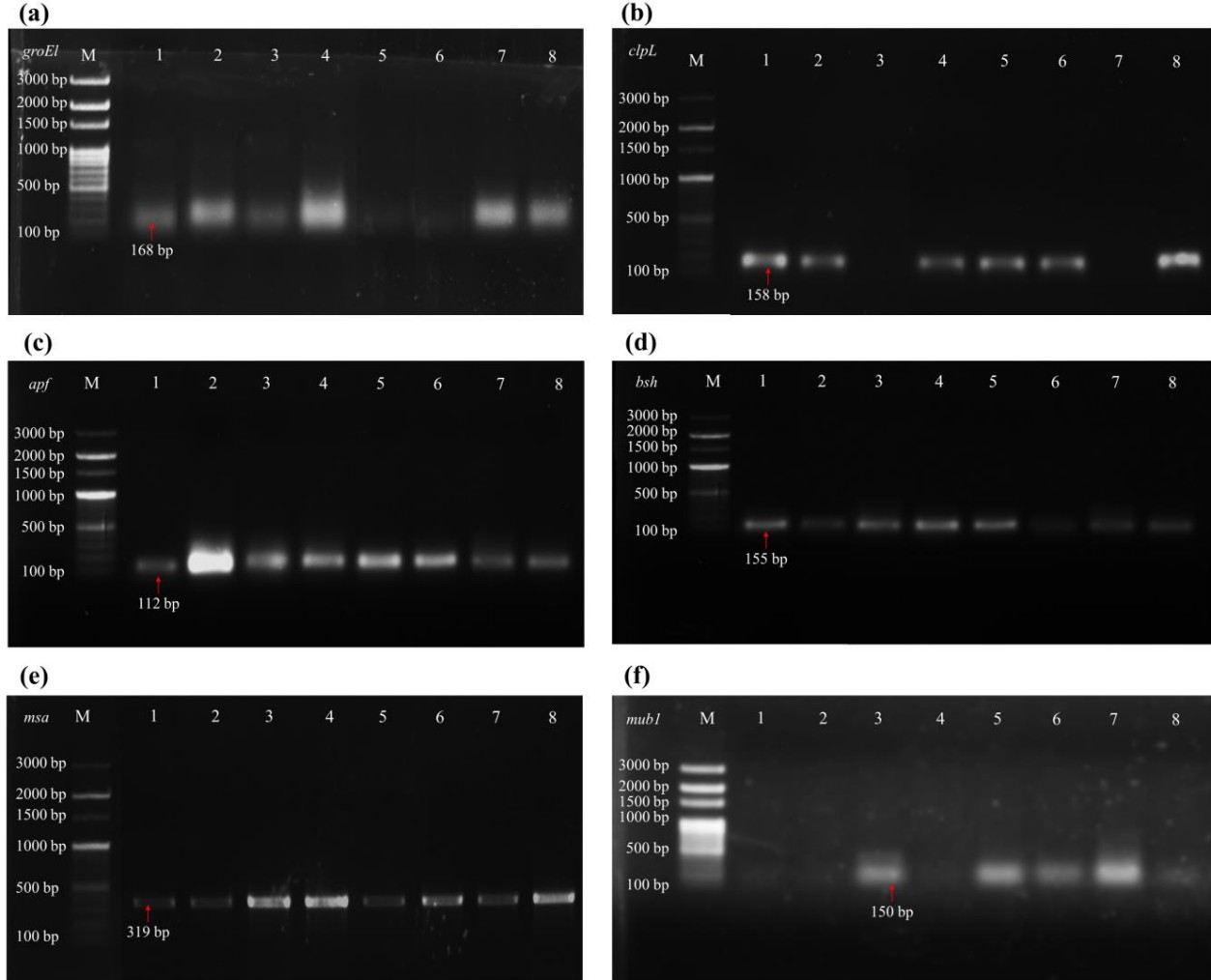

**Figure 3.** PCR amplification of marker genes associated with probiotic characteristics. Two marker

genes were associated with survival at low pH (**a**) *groEl* and (**b**) *clpL*. Two marker genes were associated with bile salt (**c**) *apf* and (**d**) *bsh*. Two marker genes were associated with adhesion/attachment (**e**) *msa* and (**f**) *mub1*. Details in the image of gel electrophoresis described as: 'M' DNA ladder (100 bp); (1) *Pediococcus acidilactici* Ki20 (*Kinema*); (2) *Enterococcus faecium* Kn19 (*Kinema*); (3) *Enterococcus faecalis* Gc21 (*Grep-chhurpi*); (4) *Pediococcus acidilactici* Ph32 (*Peha*); (5) *Enterococcus faecium* Pn11 (*Peron namsing*); (6) *Enterococcus faecalis* Pn37 (*Peron namsing*); (7) *Enterococcus lactis* Py03 (*Peruñyaan*); (8) *Enterococcus lactis* Py08 (*Peruñyaan*).

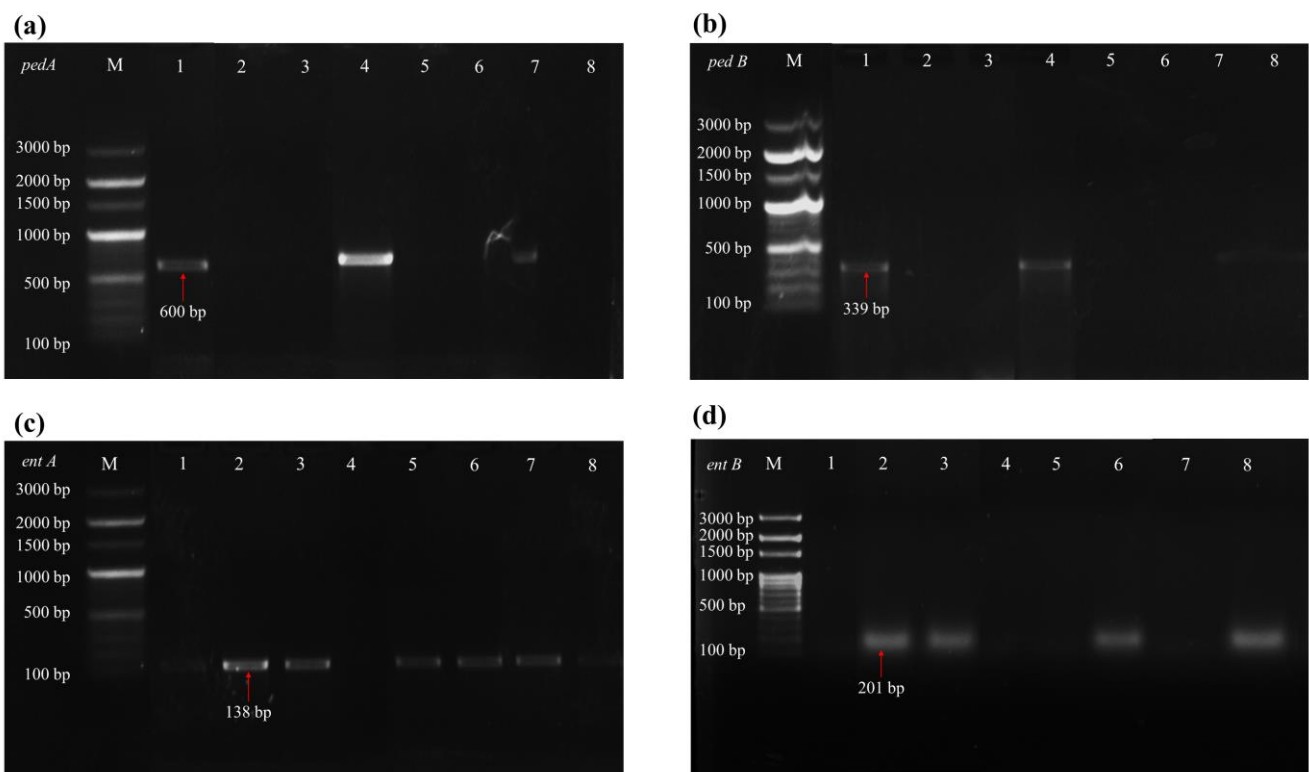

**Figure 4.** Marker genes associated with probiotic characteristics detected by PCR amplification. Detection of the gene associated with pediocin (**a**) *pedA* and (**b**) *pedB*. Marker genes were associated with enterocin (**c**) *entA* and (**d**) *entB*. Gel electrophoresis image description: 'M' DNA ladder (100 bp); (1) *Pediococcus acidilactici* Ki20 (*Kinema*); (2) *Enterococcus faecium* Kn19 (*Kinema*); (3) *Enterococcus faecalis* Gc21 (*Grep-chhurpi*); (4) *Pediococcus acidilactici* Ph32 (*Peha*); (5) *Enterococcus faecium* Pn11 (*Peron namsing*); (6) *Enterococcus faecalis* Pn37 (*Peron namsing*); (7) *Enterococcus lactis* Py03 (*Peruñyaan*); (8) *Enterococcus lactis* Py08 (*Peruñyaan*).

## 4. Discussion

*Bacillus* species, mostly *B. subtilis*, have been reported as pre-dominant bacteria from the spontaneous fermentation of soybeans into sticky with umami-flavoured foods in Asia [10,11,13,51–53], which dispense several bio-functional and health benefits [7,24,54,55]. Even some of these species, largely *Bacillus subtilis*, have been developed as the commercial starter for monoculture fermentation of soybean into some popular fermented soybean foods such as *natto* of Japan [56] and *cheonggokjang* of Korea [57]. However, the age-old traditional fermentation of soybeans is orchestrated exclusively by spontaneous fermentation, which facilitates the existence and co-existence of the complex microbial community including domains of bacteria (bacilli, lactic acid bacteria, and other non-lactis), eukaryotes (yeasts and fungi), viruses (bacteriophages), and archaea [13,52,58]. The exact role of LAB in soybean fermentation has not been extensively studied; however, their coexistence during natural fermentation has raised concern for their fermentative or non-fermentative roles in soybean fermentation. One of the bio-functional roles of LAB is probiotics, which have been reported in many other fermented foods of plant/animal-based foods. We hypothe-

sized that some LAB in Himalayan naturally fermented soybean foods may have probiotic properties, which may aid the functionality of the soybean foods.

The ability of any bacterial strain to survive in gastrointestinal transit, highly acidic and strong bile [59,60] and the adhesion property to epithelial cells [61,62] are the most important preliminary screening criteria to claim the tentative probiotic properties. Hence, we evaluated 352 bacterial isolates from the Himalayan fermented soybean foods based on their survival in low pH, bile salt, and adhesion property in terms of hydrophobicity index. Finally, eight strains of lactic acid bacteria were selected as probiotic bacteria viz. *Pediococcus acidilactici* strains Ki20 & Ph32, *Enterococcus faecium* strains Kn19 & Pn11, *E. faecalis* Gc21 & Pn37 and *E. lactis* Py03 & Py08. Species of *Enterococcus* and *Pediococcus*, isolated from other fermented soybean foods, also manifested the ability to resist low acid and bile salt, indicating their easy colonization of the human gut [63,64]. Detection of genes *groEl*, *clpL*, and *apf* in identified strains of *Enterococcus* and *Pediococcus* justify their responsibility for acid and bile tolerance [65]. Gene (*bsh*) responsible for bile salt hydrolase (BSH) usually correlated with the ability to lower serum cholesterol levels [66] was also detected in all strains of *Pediococcus* and *Enterococcus*. The microbial adhesion of LAB to hydrocarbons with a hydrophobicity index of ≥70% is being classified as highly hydrophobic strains [62]. Moreover, the results in the current study showed the hydrophobicity of ≥80% in all strains of *P. acidilactici*, *E. faecium*, *E. faecalis*, and *E. lactis*, which indicates their strong adhesion capabilities to the epithelial cells [67]. All strains showed the presence of *msa* gene, whereas only four strains of *Enterococcus* showed the presence of mucus adhesion *mub1* gene, which is responsible for cell surface adherence/attachment [68]. In addition to adhesion, auto-aggregation and co-aggregation properties of probiotic bacteria that prevent pathogens from surface colonization were also recorded. Auto-aggregation could enable self-joining association of the same species of LAB attached to the mucosal surface [69]. Co-aggregation enables the intercellular attachment/adhesion with other microorganisms bound up with the ability to interact with pathogens [34]. Therefore, the ability of LAB strains for auto-aggregation and co-aggregation is related to the adhesion capacity that has an effect on reducing mucosa colonisation by pathogenic strains [70]. The ability to colonize is an important characteristic for probiotic isolates to exhibit beneficial properties and exclude pathogenic bacteria [71]. In the present study, we also observed the resistivity to lysozyme by the species of *Pediococcus* and *Enterococcus*. Tolerance to lysozyme is the first valuable step to evaluate the potential contribution of LAB in extreme environments of gastrointestinal pathways [72]. In addition, the results recorded indicate the ability of LAB strains to inhibit the growth of pathogens, which is one of the most desirable properties of probiotic strains to exhibit antibacterial activity [73]. The antibacterial activity of LAB against pathogens and spoilage microbes could be due to the production of bacteriocin or bacteriocin-like inhibitory substances (BLIS) [74,75]. The PCR-based method also revealed the detection of pediocin genes *pedA* and *pedB*, a class of bacteriocin [76] detected in *Pediococcus acidilactici*. Pediocin produced by *Pediococcus acidilactici* might be able to create pores in the cell membrane that ultimately leads to cell death [77]. Similarly, the detection of enterocin (*entA*, *entB*), which are structural genes for enterocin Xα and Xβ, respectively [78], in *Enterococcus* species, indicated their antimicrobial spectrum. The detection of enterocin-producing genes in *Enterococcus* species might attribute to controlling the target bacteria by inducing membrane porosity that causes leakage of target cells or by interfering with the DNA replication and transcription [79].

## 5. Conclusions

The current study was conducted to evaluate some of the probiotic properties of LAB isolated from Himalayan spontaneously fermented soybean foods. *Pediococcus* spp. and *Enterococcus* spp. revealed the potential probiotic characteristics in Himalayan fermented soybean foods. This may be the first report on the role of LAB in *Bacillus*-dominated Himalayan fermented soybean food. Based on in vitro studies, several probiotic attributes were able to be exhibited by the LAB strains, which were further screened by genetic

screening that revealed the presence of some marker genes related to specific probiotic characteristics. However, further studies on whole genome analysis of probiotic strains, their safety assessment, and other functional characteristics may also be of great importance to understanding the role and contribution of probiotic LAB in soybean fermentation. *Bacillus* species were reported as the main key microbes in fermented soybean foods; however the ability of LAB strains to exhibit some functional characteristics such as organic acid production, and antibacterial properties against pathogens might attract the possible co-culture that suppresses unwanted microorganisms during fermentation. Additionally, probiotic LAB may be developed as starter(s) or co-culture(s) with *Bacillus* spp. for controlled and optimized fermentation of soybeans.

**Author Contributions:** Conceptualization, J.P.T.; methodology, P.K.; investigation, P.K.; resources, J.P.T.; data curation, J.P.T.; writing—original draft preparation, P.K.; writing—review and editing, J.P.T.; visualization, P.K.; supervision, J.P.T.; project administration, J.P.T.; funding acquisition, J.P.T. All authors have read and agreed to the published version of the manuscript.

**Funding:** This research received no external funding.

**Institutional Review Board Statement:** Not applicable.

**Informed Consent Statement:** Not applicable.

**Data Availability Statement:** The 16S rRNA gene sequences of identified lactic acid bacteria were deposited in GenBank NCBI under the accession numbers OP941712, OP941713, OP941714, OP941715, OP941716, OP941717, OP941718 and OP941719.

**Acknowledgments:** Jyoti P. Tamang is grateful to International Centre for Integrated Mountain Development (ICIMOD)—Mountain Chair for financial support.

**Conflicts of Interest:** The authors declare no conflict of interest.

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
