# Peer review of "Probiotic Properties of Lactic Acid Bacteria Isolated from the Spontaneously Fermented Soybean Foods of the Eastern Himalayas"

_fermentation, doi:10.3390/fermentation9050461_

Round 1
Reviewer 1 Report
The aim of this study is to investigate the probiotic properties of low-abundance lactic acid bacteria isolated from naturally fermented soy foods in the eastern Himalayas. A total of 352 isolated lactic acid bacteria were screened for low pH survival, bile salt tolerance and cell surface hydrophobicity. Finally, 8 strains of probiotics were selected for in vitro detection and genetic screening of probiotic properties. This study highlights the probiotic potential of low abundance LAB in naturally fermented soy foods from the eastern Himalaya and can be developed as starter or co-starter cultures for controlled and optimized fermentation of soy. There are still some problems with the manuscript that require substantial revision.
1. In the discussion section, it is necessary to further discuss the relationship between functional genes and microorganisms and the mechanism of action.
2. There are no obvious highlights in the article, which needs to be further explored.
3. In the experimental design to explore the strain in the GI environment, the survival rate of the strain under the condition "pH=3, 0.3% bile salt", which is the similar condition for digestion in human beings, was not explored. I suggest an additional experiment.
4. The effect of conducting experiments on ‘auto-aggregation’ and ‘co-aggregation’ was not explained. The significance of “auto-aggregation” and “co-aggregation” was not explained.
5. Line 92-93: I am confused of the way of colony selection was random, but not according to the phenotypic characteristics.
6. In figure 1, ‘peron-namsing’ should be consistent with the previous text.
7. I am confused of the ‘> 76% in bile salt’, most survival rates are less than 76% in the table.
8. The spelling of some words in table 3 is incomplete, please supplement the whole word. Getting information from a complex table is difficult. I suggest to supplement the bar chart to intuitively reflect the data.
9. The gene of groEl and clpL are not possessed by all species, please check.
10. Line 381-383: The reference 60 does not exist "Cell surface hydrophobicity ≥70% is considered the lowest threshold for attachment to the mucosal surface", please check. There are other errors in the reference section, please check all the references and citations.
11. The figure 3 and figure 4 should be combined into one figure. But the content of it was repeated with the heat map, I suggest to place in the attached file.
12. I am confused of ‘class IIa bacteriocin’.
Extensive editing of English language required
Author Response
Reviewers Comments
Fermentation-2362955: Probiotic properties of lactic acid bacteria isolated from the spontaneously fermented soybean foods of the Eastern Himalayas (Pynhunlang Kharnaior and Jyoti Prakash Tamang)
Comments and Suggestions for Authors
Reviewer # 1
The aim of this study is to investigate the probiotic properties of low-abundance lactic acid bacteria isolated from naturally fermented soy foods in the eastern Himalayas. A total of 352 isolated lactic acid bacteria were screened for low pH survival, bile salt tolerance and cell surface hydrophobicity. Finally, 8 strains of probiotics were selected for in vitro detection and genetic screening of probiotic properties. This study highlights the probiotic potential of low abundance LAB in naturally fermented soy foods from the eastern Himalaya and can be developed as starter or co-starter cultures for controlled and optimized fermentation of soy. There are still some problems with the manuscript that require substantial revision.
- In the discussion section, it is necessary to further discuss the relationship between functional genes and microorganisms and the mechanism of action.
Answer: As suggested by the Reviewer, we have added the information (discussion) in the revised manuscript.
- There are no obvious highlights in the article, which needs to be further explored.
Answer: As suggested by the Reviewer, we have included the highlights separately.
- In the experimental design to explore the strain in the GI environment, the survival rate of the strain under the condition "pH=3, 0.3% bile salt", which is the similar condition for digestion in human beings, was not explored. I suggest an additional experiment.
Answer: As suggested by the Reviewer, we have included the data on survival rate at low pH and bile salt (Table 3). Future study can be explored to better understand the ability of strains survival at GI environment.
- The effect of conducting experiments on ‘auto-aggregation’ and ‘co-aggregation’ was not explained. The significance of “auto-aggregation”and “co-aggregation” was not explained.
Answer: As suggested by the Reviewer, we have added in the Discussion section in revised manuscript…the ability of LAB strains for auto-aggregation and co-aggregation are related to the adhesion capacity that has an effect on reducing the mucosa colonisation by pathogenic strains [70].
- Line 92-93: I am confused of the way of colony selection was random, but not according to the phenotypic characteristics.
Answer: As suggested by the Reviewer, we have revised the sentence in the manuscript. After isolation, the LAB isolates were subjected for preliminary screening based on acid and bile tolerance followed by cell surface hydrophobicity.
- In figure 1, ‘peron-namsing’ should be consistent with the previous text.
Answer: As suggested by the Reviewer, we have corrected the typographical error of peron-namsing to peron namsing in Figure 1.
- I am confused of the ‘> 76% in bile salt’, most survival rates are less than 76% in the table.
Answer: As suggested by the Reviewer, we have corrected the error…..>66 % in bile salt.
- The spelling of some words in table 3 is incomplete, please supplement the whole word. Getting information from a complex table is difficult. I suggest to supplement the bar chart to intuitively reflect the data.
Answer: As suggested by the Reviewer, we have expanded the Table 3.
- The gene of groEland clpL are not possessed by all species, please check.
Answer: As suggested by the Reviewer, we have corrected the sentence in the discussion section.
- Line 381-383: The reference 60 does not exist "Cell surface hydrophobicity ≥70% is considered the lowest threshold for attachment to the mucosal surface", please check. There are other errors in the reference section, please check all the references and citations.
Answer: As suggested by the Reviewer, we have rephrased the sentence “The microbial adhesion of LAB to hydrocarbons with a hydrophobicity index of ≥70% is being classified as highly hydrophobic strains [62]”. We have also replaced a reference.
- The figure 3 and figure 4 should be combined into one figure. But the content of it was repeated with the heat map, I suggest to place in the attached file.
Answer: As suggested by the Reviewer, we have excluded the heatmap representation and showed only Gel electrophoresis figures (3 and 4) on probiotic marker gene detection.
- I am confused of ‘class IIa bacteriocin’.
Answer: As suggested by the Reviewer, we have corrected the typographical error
Reviewer 2 Report
The Abstract should be reorganized in way to be better representative of paper.
The context should better introduced; particularly in Introduction add some lines on advances researches of fermented foods and beverages such as:
Nazhand, et al. Ready to Use Therapeutical Beverages: Focus on Functional Beverages Containing Probiotics, Prebiotics and Synbiotics. Beverages 2020, 6, 26. https://doi.org/10.3390/beverages6020026.
Durazzo et al. Fermented food/beverage and health: current perspectives. 2022. Rendiconti Lincei. Scienze Fisiche e Naturali. 33(5):1-10. https://doi.org/10.1007/s12210-022-01093-6
A graphical scheme of study approach should be inserted.
Major details on Bioinformatics analysis should be added.
Results in Table 3 should be better described.
Considerations as well as limits and advantages of research should be added in Conclusion.
Check the format of paper in line with Journal guidelines
The linguistic revision of whole manuscript should be carried out
Author Response
Reviewer# 2
The Abstract should be reorganized in way to be better representative of paper.
Answer: As suggested by the Reviewer, we have reorganised the abstract.
The context should better introduced; particularly in Introduction add some lines on advances researches of fermented foods and beverages such as: Nazhand, et al. Ready to Use Therapeutical Beverages: Focus on Functional Beverages Containing Probiotics, Prebiotics and Synbiotics. Beverages 2020, 6, 26. https://doi.org/10.3390/beverages6020026.
Durazzo et al. Fermented food/beverage and health: current perspectives. 2022. Rendiconti Lincei. Scienze Fisiche e Naturali. 33(5):1-10. https://doi.org/10.1007/s12210-022-01093-6
A graphical scheme of study approach should be inserted.
Major details on Bioinformatics analysis should be added.
Answer: As suggested by the Reviewer, we added the sentences in Introduction “Fermented foods ….. health benefits to the consumers [1]. Several potential ….. cholesterol levels [2].
As suggested by the Reviewer, we have included the graphical abstract.
As suggested by the Reviewer, we have also rephrased the bioinformatic analysis section in the revised manuscript.
Results in Table 3 should be better described.
Answer: As suggested by the Reviewer, we have added in the revised manuscript.
Considerations as well as limits and advantages of research should be added in Conclusion.
Answer: As suggested by the Reviewer, we have added in the revised manuscript.
Check the format of paper in line with Journal guidelines
Answer: As suggested by the Reviewer, we have revised as per Journal guidelines (reference)
The linguistic revision of whole manuscript should be carried out
Answer: As suggested by the Reviewer, we have corrected the language in the revised manuscript.
Reviewer 3 Report
I believe that the reviewed article has scientific significance and is of sufficient interest to researchers.
The authors reviewed the main aspects of isolation of probiotic microorganisms from national fermented soybean-based products and confirmation of functional properties of these microorganisms.
I would recommend this manuscript for publication in the journal Fermentation after a minor revision:
I recommend reducing the font in the tables and formatting all the tables so that they become more convenient for analysis.
It is necessary to expand the conclusion, add more details of the prospects for the application of your findings and also objectives for further research.
No comment on this item
Author Response
Reviewer # 3
I believe that the reviewed article has scientific significance and is of sufficient interest to researchers. The authors reviewed the main aspects of isolation of probiotic microorganisms from national fermented soybean-based products and confirmation of functional properties of these microorganisms. I would recommend this manuscript for publication in the journal Fermentation after a minor revision:
I recommend reducing the font in the tables and formatting all the tables so that they become more convenient for analysis.
Answer: As suggested by the Reviewer, we have reduced the font size of the Tables.
It is necessary to expand the conclusion, add more details of the prospects for the application of your findings and also objectives for further research.
Answer: As suggested by the Reviewer, we have added the information in the revised manuscript.